# REMOVING DIMENSIONAL RESTRICTIONS ON COMPLEX/HYPER-COMPLEX CONVOLUTIONS

## ABSTRACT

It has been shown that the core reasons that complex and hypercomplex valued neural networks offer improvements over their real-valued counterparts is the fact that aspects of their algebra forces treating multi-dimensional data as a single entity (forced local relationship encoding) with an added benefit of reducing parameter count via weight sharing. However, both are constrained to a set number of dimensions, two for complex and four for quaternions. These observations motivate us to introduce novel vector map convolutions which capture both of these properties provided by complex/hypercomplex convolutions, while dropping the unnatural dimensionality constraints their algebra imposes. This is achieved by introducing a system that mimics the unique linear combination of input dimensions via the Hamilton product using a permutation function, as well as batch normalization and weight initialization for the system. We perform three experiments using three different network architectures to show that these novel vector map convolutions seem to capture all the benefits of complex and hyper-complex networks, such as their ability to capture internal latent relations, while avoiding the dimensionality restriction.

## 1 INTRODUCTION

While the large majority of work in the area of machine learning (ML) has been done using real-valued models, recently there has been an increase in use of complex and hyper-complex models (Trabelsi et al., 2018a; Parcollet et al., 2020). These models have been shown to handle multidimensional data more effectively and require fewer parameters than their real-valued counterparts.

For tasks with two dimensional input vectors, complex-valued neural networks (CVNNs) are a natural choice. For example in audio signal processing the magnitude and phase of the signal can be encoded as a complex number. Since CVNNs treat the magnitude and phase as a single entity, a single activation captures their relationship as opposed to real-valued networks. CVNNs have been shown to outperform or match real-valued networks, while sometimes at a lower parameter count (Trabelsi et al., 2018b; Aizenberg & Gonzalez, 2018). However, most real world data has more than two dimensions such as color channels of images or anything in the realm of 3D space.

The quaternion number system extends the complex numbers. These hyper-complex numbers are composed of one real and three imaginary components making them ideal for three or four dimensional data. Quaternion neural networks (QNNs) have enjoyed a surge in recent research and show promising results (Takahashi et al., 2017; Bayro-Corrochano et al., 2018; Gaudet & Maida, 2018; Parcollet et al., 2017a;b; 2018a;b; 2019). Quaternion networks have been shown to be effective at capturing relations within multidimensional data of four or fewer dimensions. For example the red, green, and blue color image channels for image processing networks needs to capture the cross channel relationships of these colors as they contain important information to support good generalization (Kusamichi et al., 2004; Isokawa et al., 2003). Real-valued networks treat the color channels as independent entities unlike quaternion networks. Parcollet et al. (2019) showed that a real-valued, encoder-decoder fails to reconstruct unseen color images due to it failing to capture local (color) and global (edges and shapes) features independently, while the quaternion encoder-decoder can do so. Their conclusion is that the Hamilton product of the quaternion algebra allows the quaternion network to encode the color relation since it treats the colors as a single entity. Another example is 3D spatial coordinates for robotic and human-pose estimation. Pavllo et al. (2018) showed improvement on short-term prediction on the Human3.6M dataset using a network that encoded rotations as quaternions over Euler angles.

The prevailing view is that the main reason that these complex networks outperform real-valued networks is their underlying algebra which treats the multidimensional data as a single entity (Parcollet et al., 2019). This allows the complex networks to capture the relationships between the dimensions without the trade-off of learning global features. However, only the Hamilton product seems to be needed to capture this property and the other aspects of the algebra are only imposing dimensionality constraints. Therefore, the present paper proposes: 1) to create a system that mimics

the concepts of complex and hyper-complex numbers for neural networks, which treats multidimensional input as a single entity and incorporates weight sharing, but is not constrained to certain dimensions; 2) to increase their local learning capacity by introducing a learnable parameter inside the multidimensional dot product. Our experiments herein show that these novel vector map convolutions seem to capture all the benefits of complex and hyper-complex networks, while improving their ability to capture internal latent relations, and avoiding the dimensionality restriction.

## 2 MOTIVATION FOR VECTOR MAP CONVOLUTIONS

Nearly all data used in machine learning is multidimensional and, to achieve good performance models, must both capture the local relations within the input features (Tokuda et al., 2003; Matsui et al., 2004), as well as non-local features, for example edges or shapes composed by a group of pixels. Complex and hyper-complex models have been shown to be able to both capture these local relations better than real-valued models, but also to do so at a reduced parameter count due to their weight sharing property. However, as stated earlier, these models are constrained to two or four dimensions. Below we detail the work done showing how hyper-complex models capture these local features as well as the motivation to generalize them to any number of dimensions.

Consider the most common method for representing an image, which is by using three 2D matrices where each matrix corresponds to a color channel. Traditional real-valued networks treat this input as a group of uni-dimensional elements that may be related to one another, but not only does it need to try to learn that relation, it also needs to try to learn global features such as edges and shapes. By encoding the color channels into a quaternion, each pixel is treated as a whole entity whose color components are strongly related. It has been shown that the quaternion algebra is responsible for allowing QNNs to capture these local relations. For example, Parcollet et al. (2019) showed that a real-valued, encoder-decoder fails to reconstruct unseen color images due to it failing to capture local (color) and global (edges and shapes) features independently, while the quaternion encoder-decoder can do so. Their conclusion is that the Hamilton product of the quaternion algebra allows the quaternion network to encode the color relation since it treats the colors as a single entity. The Hamilton product forces a different linear combination of the internal elements to create each output element. This is seen in Fig. 1 from Parcollet et al. (2018a), which shows how a real-valued model looks when converted to a quaternion model. Notice that the real-valued model treats local and global weights at the same level, while the quaternion model learns these local relations during the Hamilton product. This is because each output unit shares the weights and are therefore forced to discover joint correlations within the input dimensions. The weight sharing property can also be seen where each element of the weight is used four times, reducing the parameter count by a factor of four from the real-valued model.

The advantages of hyper-complex networks on multidimensional data seems clear, but what about niche cases where there are higher dimensions than four? Examples include applications where one needs to ingest extra channels of information in addition to RGB for image processing, like satellite images which have several bands. To overcome this limitation we introduce vector map convolutions, which attempt to generalize the benefits of hyper-complex networks to any number of dimensions. We also add a learnable set of parameters that modify the linear combination of internal elements to allow the model to decide how important each dimension may be in calculating others.

## 3 VECTOR MAP COMPONENTS

This section will include the work done to obtain a working vector map network. This includes the vector map convolution operation and the weight initialization used.

### 3.1 VECTOR MAP CONVOLUTION

Vector map convolutions use a similar mechanism to that of complex (Trabelsi et al., 2018b) and quaternion (Gaudet & Maida, 2018) convolutions but drops the other constraints imposed by the hyper-complex algebra. We will begin by observing the quaternion valued layer from Fig. 1. Our goal is to capture the properties of weight sharing and each output axis being composed of a linear combination of all the input axes, but for an arbitrary number of dimensions $D_{\mathrm{vm}}$.

For the derivation we will choose $D_{\mathrm{vm}} = N$. Let $V_{in}^n = [v_1, v_2, \ldots, v_n]$ be an $N$ dimensional input vector and $W^n = [w_1, w_2, \ldots, w_n]$ be an $N$ dimensional weight vector. Note that for the complex and quaternion case the output vector is a set of different linear combinations where each input vector is multiplied by each weight vector element a total of

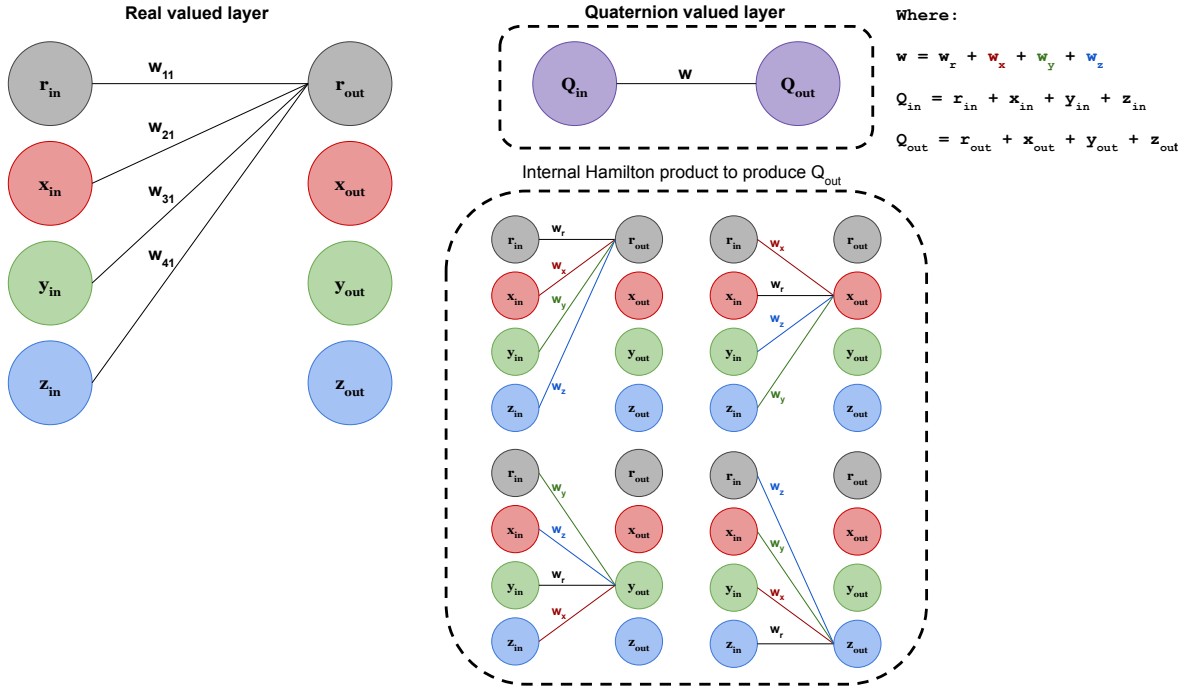

Figure 1: Illustration of the difference between a real-valued layer (left) and quaternion-valued layer (right). The quaternion's Hamilton product shows the internal relation learning ability not present in the real-valued due to sharing weights and being forced to discover joint correlations.

one time over the set. To achieve a similar result we will define a permutation function:

$$\tau(v_i) = \begin{cases} v_n & i = 1 \\ v_{i-1} & i > 1. \end{cases}$$

By applying $\tau$ to each element in $V^n$ a new vector is created that is a circular right shifted permutation of $V^n$:

$$\tau(V^n) = [v_n, v_1, v_2, \ldots, v_{n-1}].$$

Let the repeated composition of $\tau$ be denoted as $\tau^n$, then we can define the resultant output vector $V_{out}$ as:

$$V_{out}^n = \left[ W^n \cdot V_{in}^n, \tau^2(W^n) \cdot V_{in}^n, \ldots, \tau^{n-2}(W^n) \cdot V_{in}^n, \tau^{n-1}(W^n) \cdot V_{in}^n \right] \tag{1}$$

where $| \cdot |$ is the dot product of the vectors. The above gives each element of $V_{out}^n$ a unique linear combination of the elements of $V_{in}^n$ and $W^n$ since we never need to compose $\tau$ above $n-1$ times (meaning the original vector $W^n$ and any permutation only appear once). Also note that we chose right shifting permutation, but any scheme that gives a complete set of linear combinations is admissible.

The previous discussion applies to densely connected layers. The same idea is easily mapped to convolutional layers where the elements of $V_{in}^n$ and $W^n$ are matrices. To develop intuitions, the quaternion convolution operation is depicted in Fig. 3. The top of the figure shows four multichannel inputs and four multichannel kernels to be convolved. The resulting output is shown at the bottom of the figure. Each row in the middle of the figure shows the calculation of one output feature map, which is a 'convolutional' linear combination of one feature map with the four kernels (and the kernel coefficients are distinct for each row). When looking at the pattern across the rows, the weight sharing can be seen. Across the rows, any given kernel is convolved with four different feature maps. The only thing constraining the dimension to four is the coefficient values at the bottom of the figure imposed by the quaternion algebra (for more detail, see Gaudet & Maida (2018)). We hypothesize that the only thing important about the coefficient values is how they constrain the linear combinations to be independent. We also propose that the circularly shifted permutations just described generate admissible linear combinations. In this case, space permitting, Fig. 3 could be a $5 \times 5$ image, where five filters are convolved with five feature maps while the weight sharing properties are preserved. That, is there is no longer a dimensional constraint.

We also define a learnable constant defined as a matrix $\mathbf{L} \in \mathbb{R}^{D_{\text{vm}} \times D_{\text{vm}}}$:

$$
l_{i,j} = \begin{cases} 1 & i = 1 \\ 1 & i = j \\ 1 & j = (i + (i-1)) \bmod D_{\text{vm}} \\ -1 & else. \end{cases}
$$

The purpose of this matrix is to perform scalar multiplication with the input matrix, which will allow the network some control over how much the value of one axis influences another axis (the resultant axes all being different linear combinations of each axis).

With all of the above we can look at an example where $D_{\text{vm}} = 4$ so we can then compare to the quaternion convolution. Here we let the weight filter matrix $\mathbf{W} = [\mathbf{A}, \mathbf{B}, \mathbf{C}, \mathbf{D}]$ by an input vector $\mathbf{h} = [\mathbf{w}, \mathbf{x}, \mathbf{y}, \mathbf{z}]$

$$
\begin{bmatrix} D_1(\mathbf{W} * \mathbf{h}) \\ D_2(\mathbf{W} * \mathbf{h}) \\ D_3(\mathbf{W} * \mathbf{h}) \\ D_4(\mathbf{W} * \mathbf{h}) \end{bmatrix} = \mathbf{L} \odot \begin{bmatrix} \mathbf{A} & \mathbf{B} & \mathbf{C} & \mathbf{D} \\ \mathbf{D} & \mathbf{A} & \mathbf{B} & \mathbf{C} \\ \mathbf{C} & \mathbf{D} & \mathbf{A} & \mathbf{B} \\ \mathbf{B} & \mathbf{C} & \mathbf{D} & \mathbf{A} \end{bmatrix} * \begin{bmatrix} \mathbf{w} \\ \mathbf{x} \\ \mathbf{y} \\ \mathbf{z} \end{bmatrix} \tag{2}
$$

where

$$
\mathbf{L} = \begin{bmatrix} 1 & 1 & 1 & 1 \\ -1 & 1 & 1 & -1 \\ 1 & -1 & 1 & -1 \\ -1 & -1 & 1 & 1 \end{bmatrix} \tag{3}
$$

The operator $| \odot |$ denotes element-wise multiply. The sixteen parameters within $\mathbf{L}$ are the initial values. They are otherwise unconstrained scalars and intended to be learnable. Thus, the vector map convolution is a generalization of complex, quaternion, or octonion convolution as the case may be, but it also drops the constraints imposed by the associated hyper-complex algebra.

For comparison the result of convolving a quaternion filter matrix $\mathbf{W} = \mathbf{A} + i\mathbf{B} + j\mathbf{C} + k\mathbf{D}$ by a quaternion vector $\mathbf{h} = \mathbf{w} + i\mathbf{x} + j\mathbf{y} + k\mathbf{z}$ is another quaternion,

$$
\begin{bmatrix} \mathscr{R}(\mathbf{W} * \mathbf{h}) \\ \mathscr{I}(\mathbf{W} * \mathbf{h}) \\ \mathscr{J}(\mathbf{W} * \mathbf{h}) \\ \mathscr{K}(\mathbf{W} * \mathbf{h}) \end{bmatrix} = \begin{bmatrix} \mathbf{A} & -\mathbf{B} & -\mathbf{C} & -\mathbf{D} \\ \mathbf{B} & \mathbf{A} & -\mathbf{D} & \mathbf{C} \\ \mathbf{C} & \mathbf{D} & \mathbf{A} & -\mathbf{B} \\ \mathbf{D} & -\mathbf{C} & \mathbf{B} & \mathbf{A} \end{bmatrix} * \begin{bmatrix} \mathbf{w} \\ \mathbf{x} \\ \mathbf{y} \\ \mathbf{z} \end{bmatrix}, \tag{4}
$$

where $\mathbf{A}$, $\mathbf{B}$, $\mathbf{C}$, and $\mathbf{D}$ are real-valued matrices and $\mathbf{w}$, $\mathbf{x}$, $\mathbf{y}$, and $\mathbf{z}$ are real-valued vectors. See Fig. 3 in the Appendix for a visualization of the above operation. More explanation is given in Gaudet & Maida (2018).

The question arises whether the empirical improvements observed in the use of complex and quaternion deep networks are best explained by the full structure of the hyper/complex algebra, or whether the weight sharing underlying the generalized convolution is responsible for the improvement.

### 3.2 Vector Map Weight Initialization

Proper initialization of the weights has been shown to be vital to convergence of deep networks. The weight initialization for vector map networks uses the same procedure seen in both deep complex networks (Trabelsi et al., 2018b) and deep quaternion networks (Gaudet & Maida, 2018). In both cases, the expected value of $|W|^2$ is needed to calculate the variance:

$$
\mathbb{E}[|W|^2] = \int_{-\infty}^{\infty} x^2 f(x) \, dx \tag{5}
$$

where $f(x)$ is a multidimensional independent normal distribution where the number of degrees of freedom is two for complex and four for hyper-complex. Solving Eq. 5 gives $2\sigma^2$ for complex and $4\sigma^2$ for quaternions. Indeed, when solving Eq. 5 for a multidimensional independent normal distribution where the number of degrees of freedom is $D_{\text{vm}}$, the solution will equal $D_{\text{vm}}\sigma^2$. Therefore, in order to respect the Glorot & Bengio (2010) criteria, the variance would be equal to :

$$
\sigma = \sqrt{\frac{2}{D_{\text{vm}}(n_{in} + n_{out})}} \tag{6}
$$

and in order to respect the He et al. (2015b) criteria, the variance would be equal to:

$$\sigma = \sqrt{\frac{2}{D_{\mathrm{vm}} n_{in}}}. \tag{7}$$

This is used alongside a vector of dimension $D_{\mathrm{vm}}$ that is generated following a uniform distribution in the interval $[0, 1]$ and then normalized.

### 3.3 Vector Map Weight Batch Normalization

Batch normalization (Ioffe & Szegedy, 2015) is used by the vast majority of all deep networks to stabilize training by keeping activations of the network at zero mean and unit variance. A formulation for complex batch normalization is given by Trabelsi et al. (2018b) in which a whitening approach is used to give equal variance to the two components. This method uses the inverse of the covariance matrix, which for the complex case is $2 \times 2$ and has an analytical solution. Similarly, Gaudet & Maida (2018) applied the same method for quaternion batch normalization, but since the $4 \times 4$ covariance matrix does not have an analytical solution for the inverse the Cholesky decomposition is used instead for whitening. We adopt the Cholesky whitening method here since we want to handle any number of dimensions giving us our vector map batch normalization:

$$BN(\tilde{x}) = \gamma \tilde{x} + \beta \tag{8}$$

where $\tilde{x}$ is the 0-centered data given by

$$\tilde{x} = \mathbf{W}(\mathbf{x} - \mathbb{E}[\mathbf{x}])$$

and where $\mathbf{W}$ is the Cholesky decomposition of the covariance matrix of the component dimensions. The scaling parameter $\gamma$ is a $D_{\mathrm{vm}} \times D_{\mathrm{vm}}$ matrix initialized to 0 everywhere except the diagonal which is set to $1/\sqrt{D_{\mathrm{vm}}}$ in order to obtain a modulus of 1 for the variance of the normalized value. The shifting parameter $\beta$ is a vector of length $D_{\mathrm{vm}}$ that is initialized to 0. It is worth noting that the Cholesky decomposition is expensive and scales $\mathbf{O}(n^2)$ with the number of dimensions of the input.

## 4 Experiments and Results

We perform three sets of experiments designed to see baseline performance, compare against some known quaternion results, and to test extreme cases of dimensionality in the data. This is done by simple classification on CIFAR data using different size ResNet models for real, quaternion, and vector map. The second experiment replicates the results of colorizing images using a convolutional auto-encoder from Parcollet et al. (2019), but using vector map convolution layers. Lastly, the DSTL Satellite segmentation challenge from Kaggle (Kaggle, 2014) is used to demonstrate the high parameter count reduction when vector map layers are used for high dimensional data. We purposefully choose commonly used architectures for comparison since we are not aiming for state-of-the-art results, but rather to show we maintain the benefits of the Hamilton product while dropping dimensionality constraints.

### 4.1 CIFAR Classification using ResNet

#### 4.1.1 CIFAR Methods

These experiments use the ResNet architecture to perform simple image classification using CIFAR-10 and CIFAR-100 datasets (Krizhevsky & Hinton, 2009). The CIFAR datasets are $32 \times 32$ color images of 10 and 100 classes respectively. Each image contains only one class and labels are provided. Since the CIFAR images are RGB, we use $D_{\mathrm{vm}} = 3$ for all the experiments.

For the architecture we use different Residual Networks taken directly from the original paper He et al. (2015a). We ran direct comparisons between real-valued, quaternion, and vector map networks on three different sizes: ReNet18, ResNet34, and ResNet50. The only change from the real-valued to vector map networks is that the number of filters at each layer is changed such that the parameter count is roughly the same as the real-valued network. The batch size was 64 for all experiments. All other hyper-parameters remain the default from the original real-valued implementation.

All use the backpropagation algorithm with Stochastic Gradient Descent with Nesterov momentum set at 0.9. The norm of the gradients were clipped to 1 and a custom learning rate scheduler was used. The learning rate was initially set to 0.01 for the first 10 epochs and then set to 0.1 from epoch 11-100 and then cut by a factor of 10 at epochs 120 and 150.

Table 1: Percent error for classification on CIFAR-10 and CIFAR-100 where Params is the total number of parameters.

| Architecture | Params | CIFAR-10 | CIFAR-100 |
|---|---|---|---|
| ResNet18 Real | 11,173,962 | 5.92 | 27.81 |
| ResNet18 Quaternion | 8,569,242 | 5.92 | 28.77 |
| ResNet18 Vector Map | 7,376,320 | 6.05 | 27.18 |
| ResNet34 Real | 21,282,122 | 5.73 | 28.18 |
| ResNet34 Quaternion | 16,315,610 | 5.73 | 27.24 |
| ResNet34 Vector Map | 14,044,960 | 5.55 | 25.88 |
| ResNet50 Real | 23,520,842 | 6.10 | 27.40 |
| ResNet50 Quaternion | 18,080,282 | 6.10 | 27.32 |
| ResNet50 Vector Map | 15,559,120 | 5.72 | 25.16 |

### 4.1.2 CIFAR RESULTS

The results are shown in Table 1 for all experiments. The vector map network's final accuracy outperforms the other models in all cases except one and the accuracy rises faster than both the real and quaternion valued networks. This may be due to the ability to control the relationships of each color channel in the convolution operation, while the quaternion is stuck to its set algebra, and the real is not combining the color channels in a similar fashion to either. For future insight into the last statement we investigated the **L** values after training and observed that they became normally distributed around -1 and 1, meaning they did in fact change some of the relations.

### 4.2 CONVOLUTIONAL AUTO-ENCODER FOR COLOR RECOVERY

### 4.2.1 CAE METHODS

This experiment originally was to explore the power of quaternion networks over real-valued by investigating the impact the Hamilton product had on reconstructing color images from gray-scale only training (Parcollet et al., 2019). A convolutional encoder-decoder (CAE) was used to test color image reconstruction. We performed the exact same experiment using quaternions, but also two experiments using vector map layers with $D_{vm} = 3$, one with **L** frozen for training and one allowing **L** to update as normal. This way we can test if we mimic the quaternion results with three dimensions by capturing the important components of treating the input dimensions as a single entity with three dimensions and also seeing the potential gain of learning **L**. The identical architecture is used, two convolutional encoding layers followed by two transposed convolutional decoding layers.

A single image is first chosen, then converted to gray-scale using the function $GS(p_{x,y})$, where $p_{x,y}$ is the color pixel at location $x, y$. The gray value is concatenated three times for each pixel to create the input to the vector map CNN. We used the exact same model architecture, but since the output feature maps is three times larger in the vector map model we reduce their size to 10 and 20. The kernel size and strides are 3 and 2 for all layers. The model is trained for 3000 epochs using the Adam optimizer (Kingma & Ba, 2014) using a learning rate of $5e^{-4}$. The weights are initialized following the above scheme in Section 3.2 and the hardtanh (Collobert, 2004) activation function is used in both the convolutional and transposed convolutional layers.

### 4.2.2 CAE RESULTS AND DISCUSSIONS

The results can be seen in Fig. 2 where one can see the vector map CAE was able to correctly produce color images like the quaternion CAE. Similar to the quaternion CAE, the vector map CAE appears to learn to preserve the internal relationship between the pixels similar to the Hamilton. The reconstructed images were also evaluated numerically using the peak signal to noise ratio (PSNR) (Turaga et al., 2004) and the structural similarity (SSIM) (Wang et al., 2004). These evaluations appear in Table 2.

The main goal of this experiment was to test if there exists a property of the quaternion structure that may have not been captured with the attempted generalization of vector map convolutions. Since both vector map networks perform similarly to the quaternion network it appears that the way the vector map rules are constructed enable it to capture the essence of the Hamilton product for any dimension size $D_{vm}$ and the additional aspects of the algebraic structure are not important. Since the $D_{vm} = 3$ model matched the quaternion performance, we have shown that the same performance can be achieved with fewer parameters.

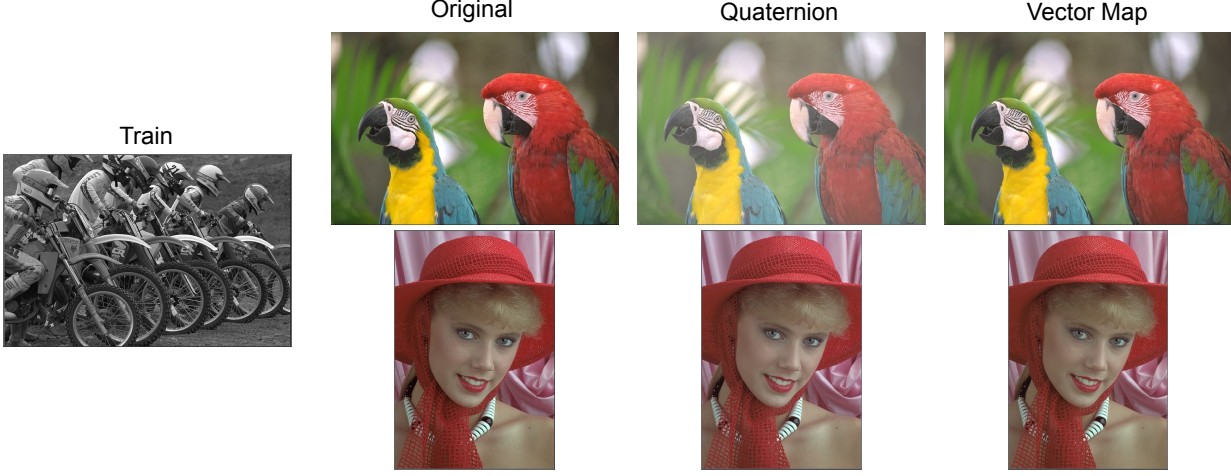

Figure 2: Grey-scale to color results on two KODAK images for a quaternion CAE and a vector map CAE.

Table 2: PSNR and SSIM results for vector map CAE with both frozen **L** and non-frozen **L** compared to quaternion CAE.

| **Image** | **Quaternion** | | $D_{\mathrm{vm}} = 3$ **frozen L** | | $D_{\mathrm{vm}} = 3$ **learned L** | |
| | **PSNR** | **SSIM** | **PSNR** | **SSIM** | **PSNR** | **SSIM** |
| --- | --- | --- | --- | --- | --- | --- |
| kodim23 | 31.68dB | 0.961 | 32.03dB | 0.973 | 32.52dB | 0.981 |
| kodim04 | 28.57dB | 0.954 | 29.18dB | 0.962 | 30.55dB | 0.972 |

### 4.3 DSTL USING UNET

The DSTL Satellite Imagery Feature Detection challenge was run on Kaggle (Kaggle, 2014) where the goal was to segment, or classify at a pixel level, 10 classes from 1km x 1km satellite images in both 3-band and 16-band formats. Since satellite images have many more bands of information than standard RGB, it makes it a good use case for vector map convolutions. We run experiments using the full bands on both real-valued and vector map networks. Note the top scoring solutions used an ensemble of several models, some trained on only particular classes, while we are doing end-to-end training on a basic model to do a comparison of real-valued to vector map networks.

### 4.4 DSTL METHODS

Both models use a standard UNet base as described in the original paper Ronneberger et al. (2015). We use the entire 16-band format as input, simply concatenating them into one input to the models. For the vector map network we choose $D_{\mathrm{vm}} = 16$ to treat the entire 16-band input as a single entity. The real-valued model has a starting filter count of 32, while the vector map has a starting filter count of 96. The images are very large so we sample from them in sizes of 82 x 82 pixels, but only use the center 64 x 64 pixels for the prediction. For training, the batch size is set to 32, the learning rate is 1e-3, and we decay the learning rate by a factor of 10 every 25 epochs during a total of 75 epochs. The cost function used is Intersection Over union (IOU), also known as the Jaccard Index, which is defined as:

$$IOU = TP/(TP + FP + FN) \tag{9}$$

where $TP$ is the number of true positives, $FP$ is the number of false positives, and $FN$ is the number of false negatives.

Table 3: IOU score on DSTL Satellite segmentation challenge for real-valued and vector map UNet models where Params is the total number of parameters.

| Architecture | Params | IOU Score |
|---|---|---|
| UNet Real | 7,855,434 | 0.427 |
| UNet Vector Map | 5,910,442 | 0.436 |

Some of the classes of the data set are only in a couple of images. For this reason, we train on all available images, but hold out random 400 x 400 chunks of the original images. We use the same seed for both the real-valued and vector map runs.

### 4.5 DSTL RESULTS

The results are shown in Table 3 where one can see that for a lower parameter budget, the vector map achieved better segmentation performance. The loss curves (not shown) display near identical shapes, but the vector map starts at a higher score and ends at a higher score. Some of the features, like the vegetation and water, stand out more distinctly in the non-RGB bands of information and the vector map seems to have captured this more accurately. The main goal was to show that the vector map convolutions could handle a large number of input dimensions and potentially better capture how the channels relate to one another, which was successful.

## 5 CONCLUSIONS

This paper proposes vector map convolutions to generalize the beneficial properties of convolutional complex and hyper/complex networks to any number of dimensions. We also introduce a new learnable parameter to modify the linear combination of internal features.

The first set of experiments compares performance of vector map convolutions against real-valued networks in three different sized ResNet models on CIFAR datasets. They demonstrate that vector map convolution networks have similar accuracy at a reduced parameter count effectively mimicking hyper-complex networks while consuming fewer resources. We also investigate the distribution of the final values of $\mathbf{L}$, the linear combination terms, and see that they also tend to stay around the value they were initialized to, suggesting that balanced magnitude linear combinations are near optimal.

We further investigated if vector map convolutions effectively mimic quaternion convolution in its ability to capture color features more effectively with the Hamilton Product with image color reconstruction tests. The vector map convolution model not only can reconstruct color like the quaternion CAE, but it performs better as indicated by PSNR and SSIM measures. This shows that other aspects of the quaternion algebra are not relevant to this task and suggests that vector map convolutions could effectively capture the internal relation of any dimension input for different data types.

The final experiment tested the ability of vector map convolutions to perform well on very high dimensional input. We compared a real-valued model against a vector map model in the Kaggle DSTL satellite segmentation challenge dataset, which has 12 channels of image information and contains 10 classes. The vector map model was built with $D_{\mathrm{vm}} = 12$ and not only had fewer learnable parameters than the real-valued model, it achieved a higher Jaccord score and learned at a faster rate. This establishes advantage of vector map convolutions in higher dimensions.

This set of experiments has shown that vector map convolutions appear to not only capture all the benefits of complex/hyper-complex convolutions, but can outperform them using a smaller parameter budget while also being free from their dimensional constraints.

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

# A APPENDIX

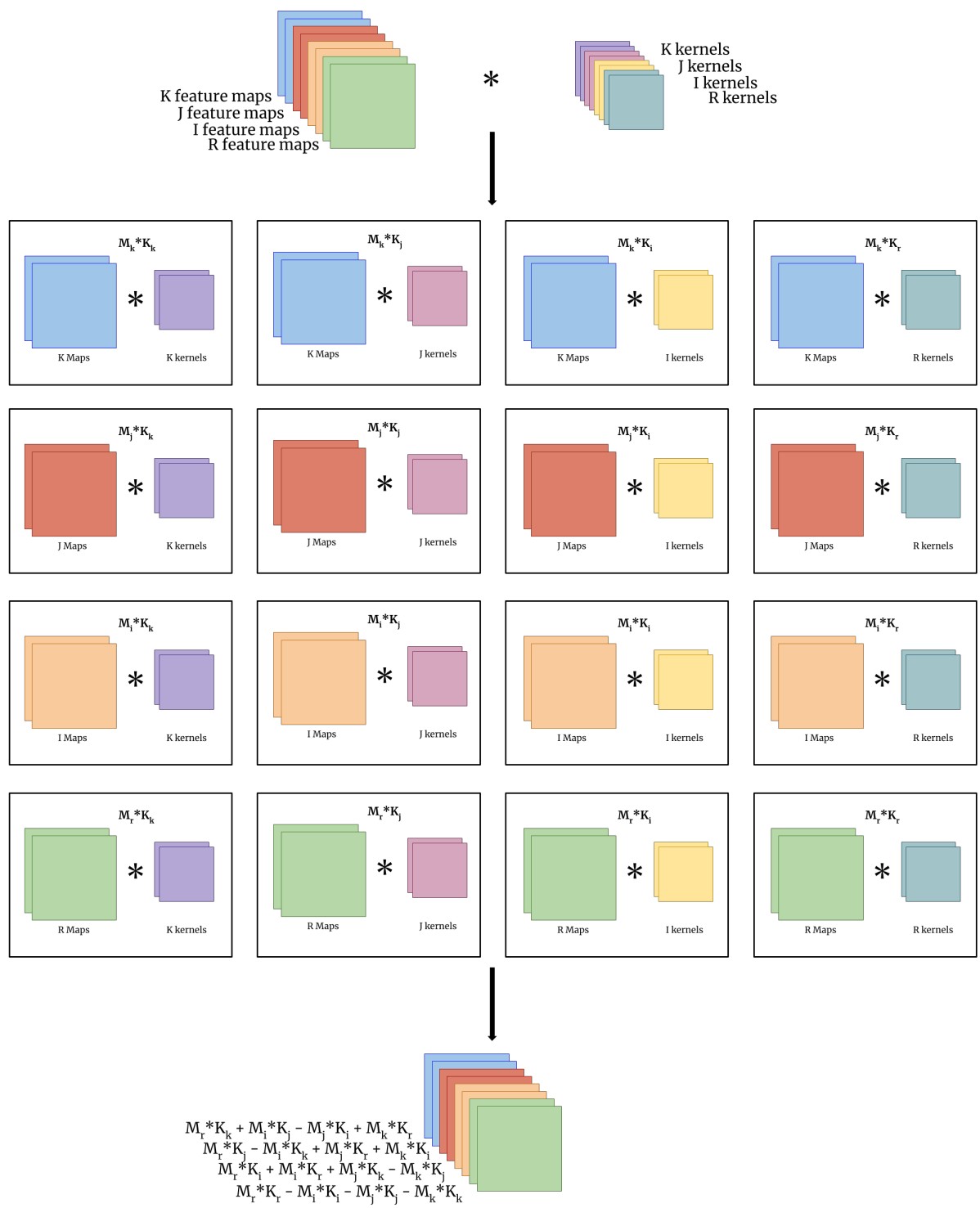

Figure 3: An illustration of quaternion convolution.

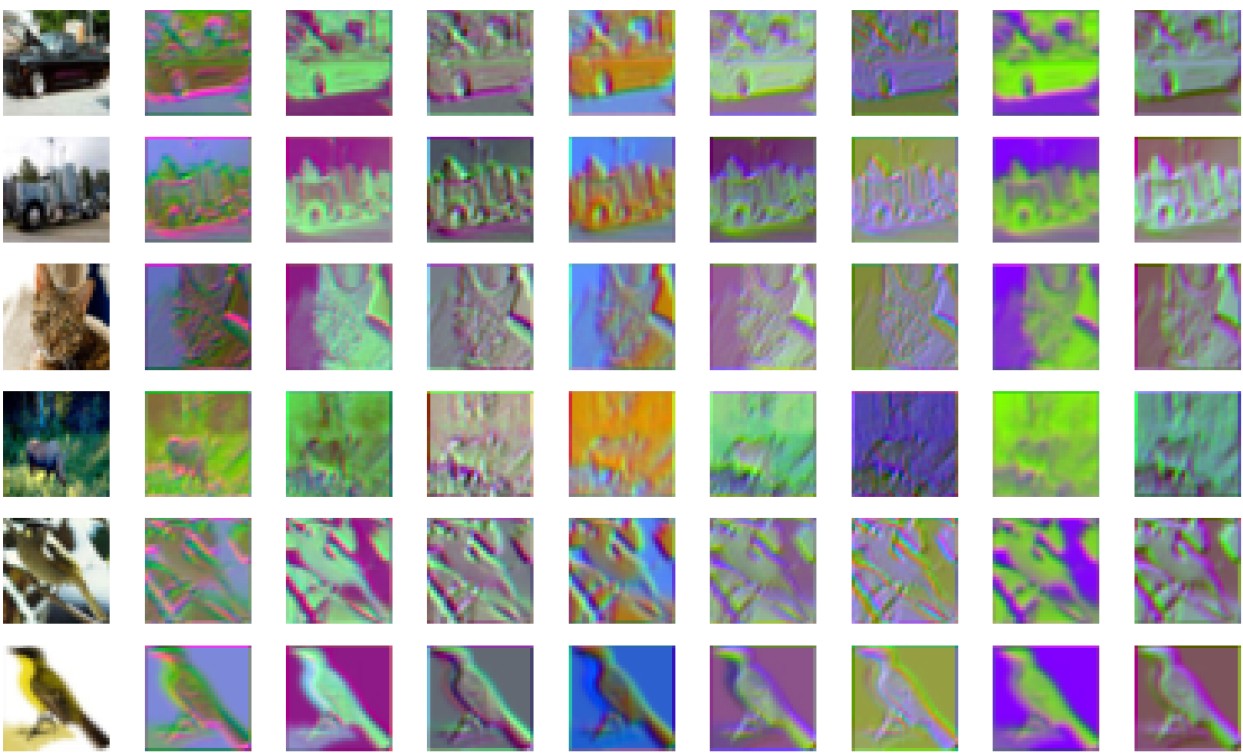

Figure 4: Randomly selected feature vector maps from the first convolution layer after training. Each row is a different image, where the first column are the original input images.

