# OpenReview forum: "Removing Dimensional Restrictions on Complex/Hyper-complex Convolutions"
_ICLR.cc/2021/Conference — Reject_

### Official Review · AnonReviewer3 · 2020-10-20
**Novel formulation to capture weight sharing and dimension relation of complex/hypercomplex networks, without the dimensionality constraints.**

**Rating:** 5
**Confidence:** 5

**Review:**

The paper proposes to explain the reason why the quaternion convolutional process reaches better performances than  neural networks with real values. this work is mostly based on previous results obtained in NLP conferences on QNN.

The idea is worth of interest but the results obtained during the experiments (from other previously already published papers) are not convincing and more efforts have to be provided in the theoretical side. For exemple, why convolution process map in a more efficient way real features in quaternion (color) space? How the Hamilton product emphasis the hidden relations between these features?

If this is mainly the fact of the Hamilton product, the authors have to give some words on the quaternion space obtained from the real valued feature map. Is this space invariant in regards to the real-valued input observed (even the batch may gives some basic combination of these input features at each tilmestep)?

---

> ### Author Response · Authors · 2020-11-23
> **Response to Reviewer 3**
>
> Our results show that there is a misconception about why the quaternion algebra has better performance on the tasks we studied in our paper. Our experiments show that only the forced local relationships (each output dimension being a unique linear combination of all dimensions and a shared weight set) is needed, while the other aspects of the algebra are not relevant. This allows us to drop dimensionality constraints imposed by the algebras.
>
> The Hamilton Product built into the quaternion algebra is what gives it the ability to better capture relations among the colors (RGB) compared to the real-valued model. This is because each output unit shares the weights and are therefore forced to discover joint correlations within the input dimensions, even for single pixels across several channels.
>
> The CAE colorization experiments reveal that we captured this property of the quaternion networks, while dropping the dimensionality constraints imposed by the algebras.

---

### Official Review · AnonReviewer4 · 2020-10-28
**Generalizing Complex/Hyper-complex Convolutions to Vector Map Convolutions**

**Rating:** 4
**Confidence:** 4

**Review:**

Paper summary

The authors propose a generalization of complex and quaternion algebra for use in convolutional neural networks, so called vector map convolutions. It is proposed that using these operations in a convolutional kernel will result in higher parameter efficiency for the same accuracy. The authors derive the functional formula for the vector map by generalizing the quaternion algebra proposed in a previous paper in the literature.

---------------------------------------------------------------------------------------------------------------------------
Positives and negatives

+ The approach is very interesting from a theoretical perspective, and the derivation proposed is well motivated and sufficiently explained in the methods section.

+ The figures are well constructed and illustrate the point very concisely.

- The major problem with this paper is that it severely goes over the 8 page limit, which is hidden by putting a lot of figures that are crucial to understanding the main text in the appendix. I believe this is a violation of the submission rules.

- The paper does not do a good enough job of discussing related approaches that also reduce parameter efficiency (e.g. separable convolutions, dynamic convolutions, stand-alone self-attention) or that mix channels in a similar spirit (e.g. ICLR 2020 “multiplicative interactions and where to find them”)

- The results are not convincing. For the satellite data case, the authors show that their proposal achieves a slightly lower Jaccard Score with fewer parameters, but it’s not clear at all what is the Jaccard Score or what its scale is. So I have no idea how a “UNet Real” Architecture with 5.9M parameters would compare with the proposed Vector Map Unet.

- In the case of CIFAR-10 and CIFAR-100 the architecture does seem to give some improvements in parameter efficiency. But how does it compare to previous work in parameter efficiency as discussed above?

---------------------------------------------------------------------------------------------------------------------------
Recommendation

With the paper in its current state I have to recommend a strong reject because it seems to be violating the conference guidelines by putting several crucial images beyond the 8 page limit. If this issue is addressed, the recommendation can be reviewed. In that case, I would still recommend more work in the experiments section (More comparisons for the image classification task, better explanation and comparisons for satellite task).

---------------------------------------------------------------------------------------------------------------------------
Questions

What is the Jaccard Score?
While the networks are shown to be more parameter efficient, what about the FLOPS of the resulting networks? Are we effectively trading memory for compute?

---------------------------------------------------------------------------------------------------------------------------
Feedback (not related to the score)

My suggestion to the authors to gain some space would be to completely remove section 4.2 or move to appendix, and use that space for the actually important figures. Not only is current figure 2 occupying a lot of space while making a weak argument, the whole section does not help in understanding either why Vector Map convolutions work, nor persuade the reader that they are empirically better (in fact the PSNR results look quite a bit worse).
In general I would recommend the authors to tighten their writing quite a bit. For example the authors repeat quite a lot of motivation for Hamiltonian algebra. While it’s useful to provide some context in a paper, the authors could summarize that information a bit better so that the reader can get to the actual contribution, the Vector Map convolution faster. On the other hand, it might be worthwhile spending a bit more time explaining  the DSTL challenge, even with a figure, since most readers will not be familiar with it.

---

> ### Author Response · Authors · 2020-11-23
> **Response to Reviewer 4**
>
> - In regards to discussing related approaches that also reduce parameter overhead, we revised the paper to emphasize that our main goal as to capture the aspects of the algebra that we proposed are important, namely the forced local relations (each output dimension being a unique linear combination of all dimensions and a shared weight set). This should clarify that we are only getting parameter count reduction of an added benefit, but the main goal is to capture that each output unit shares the weights and are therefore forced to discover joint correlations within the input dimensions. The reviewer's suggested experiments comparing other models have merit if one is trying to establish SOTA performance, however that is not our main point and we reserve that for future work.
> - We feel the overall goal clarification of trying to capture complex/hyper-complex algebras forced local relations may paint the experiments in a different light. The CIFAR work shows a simple comparison against work previously done with complex and hyper-complex CNNs. The CAE experiments show strong evidence we did capture the important piece of the compared quaternion results, while dropping the dimensionality constraints imposed by the algebra. To remove the confusion about the Jaccard Score we pointed out that it is the Intersection-Over-Union (IOU). We do not understand the concern with comparing the real-valued U-Net model to the Vector Map U-Net because the data presented in Table 3 did show the results of the comparison.
>
> ---
> Addressing Questions and Feedback:
>
> We added clarification about the increased computation of both quaternion and vector map batch-norm. In a way you are trading memory for compute, but the quaternion and vector map models have higher accuracy.
>
> Thank you for the space reduction feedback, we have removed unneeded figures.
>
> We have also fixed the numbers in Table 2 (PSNR results).

---

> > ### Comment · AnonReviewer4 · 2020-11-25
> > **Improvement**
> >
> > Thanks to the authors for their response. The paper is improved from last time and I had an easier time reading it this time around.
> >
> > The authors point is that this network architecture is being compared to the previously published Hamiltonian network and that it provides the same or better performance with fewer limitations. I accept that this point has been shown in the paper.
> >
> > However, I believe it is crucial to compare this method with other parameter reduction techniques, regardless of whether SOTA performance is established or not (I agree that it is not a necessary benchmark to warrant publication). Otherwise the paper is only relevant within the very narrow scope of the previously published paper on Hamiltonian networks, and would be better suited as a workshop paper.

---

### Official Review · AnonReviewer2 · 2020-10-28
**Potentially interesting idea to design networks with better weight-sharing; paper needs more overall polishing**

**Rating:** 4
**Confidence:** 2

**Review:**

Summary: The purpose of this paper is to analyze the reasons why complex/hyper-complex neural networks yield performance improvements (especially pertaining to generalization). The authors argue that the underlying algebraic structure of complex/hyper-complex coordinate systems enable greater weight-sharing compared to usual real-valued networks. Inspired by this, the authors also propose the idea of vector map convolutions which captures the aforementioned properties but at the same time, are not subjected to the dimensionality constraints. Several empirical studies are also provided to demonstrate the potential effectiveness of vector map convolutions.

I have to first acknowledge that I have not very familiar with much of the background literature on complex and quarternion networks. Also, for the experiments in Sections 4.2 and 4.3, I am not familiar with any of these data-sets as well as the evaluation measures (Jaccard score, etc.). Hence, I am not too confident in my assessment of this paper.

Detailed Comments/Questions:

- I would prefer a clearer and more comprehensive way of introducing the background work and how it leads to the intuition of vector map convolutions. It would be much better if the authors could provide exact mathematical descriptions of usual complex and quarternion networks (esp. as this paper is < 8 pages right now) and how vector map convolutions are the natural generalization- I don't see this as obvious based on looking at the equations in Sections 3.1

- I assume that Equation (1) and the discussion above it is the definition of the vector map convolutions? So only right-shift permutations are used and not other permutations (i.e., a subgroup instead of the full symmetric group S_n?)

- I think that the results for CIFAR-10/100 are potentially interesting as they show that the proposed approach achieves competitive performance with the standard approach while reducing parameter count significantly. However, there's far too little details given about the experimental setup; for example, what are the hyper-parameters used (Batch-size, learning rate, optimizer choices, batch-norm, dropout, etc.) and are they used in a consistent way when comparing between real, quarternion, and vector map?

===============
Post-rebuttal:

I would like to thank the authors for their rebuttal and addressing some of my concerns. However, after reading the updated manuscript as well as the other reviews, I decide to maintain my current ranting. I would still like to see more rigour in the paper: more of the mathematics need to be fleshed out and how the proposed approach compares with the existing works in complex/quarternion networks in a more mathematical way.

---

> ### Author Response · Authors · 2020-11-23
> **Response to Reviewer 2**
>
> - We revised the paper to emphasize that our main goal as to capture the aspects of the algebra that we proposed are important, namely the forced local relations. Our experiments reveal that other aspects of the complex/hyper-complex algebras are immaterial. This should make it more clear that the goal of the derivation is to capture the property that each output dimension is a unique linear combination of all input dimensions and a shared weight set.
> - We chose right shifting permutations for convenience, but any scheme that gives a complete set of linear combinations is admissible. We have added sentences after equation 1 to make this clear.
> - We have added several sentences in the relevant methods sections clarifying the hyper-parameters for the experiments and to clarify that nothing special is done for our model (it uses the exact same parameters as the real-valued), in comparison to the quaternion network. We note that batch-norm for both quaternions and vector map involve a special calculation of matrix inverse described in Section 3.3 that is different than real-valued networks.

---

### Official Review · AnonReviewer1 · 2020-10-29
**Good direction, but theory and experiments need improvement**

**Rating:** 6
**Confidence:** 4

**Review:**

# Summary

The paper proposes a method that generalizes complex and quaternion networks. The proposed method can work with vectors of any dimension. The benchmarks are conducted on CIFAR-10, CIFAR-100 and satellite imaging data DSTL. The paper also conducts an experiment on recoloration.

# Quality

While I like the general idea to explore the methods of using complex and hyper-complex representations, I found that this work is weak both in terms of the theoretical and experimental research.

The theoretical part of the paper proposes a method to multiply n-dimensional vectors similarly to complex and quaternion numbers. This method is not a generalization of the former two (as seen in Eq 2, 4). I am concerned that the paper does not discuss the properties of the operation. Such numbers do not form an algebra and they are not commutative. While quaternions are neither commutative they exhibit useful properties, like associations with rotations.

Another important aspect missing in the theoretical part of the paper is how to define the batch-norm.

The experimental part also looks weak. I am not convinced about the significance of CIFAR results. There is also no "apple to apple" comparison for the networks with the same amount of parameters. The results reported in Table 2 contradict the claim that the proposed method outperforms the quaternions.

The DSTL results look significant. I think, this result demonstrates the strength of the proposed method. Therefore I encourage to extend this section to give more details on this experiment. Even though, the result is not SOTA here, the previous results should be mentioned in the table (Kaggle winner entry is ~0.49).

# Clarity

The paper is clearly written and easy to follow. The math is explained well and helps to build intuition about the proposed method. One point I found unclear if the L is shared or learned separately for each layer. This seems to be an important difference from complex and quaternion numbers. The paper would benefit from discussion of parameter L and ablation experiments on its importance.

# Originality

The paper is original.

# Significance

The paper is significant for the community. There are plenty of applications where we need to encode multidimensional data. While I do not believe that this work is a clear cut solution for such applications, it is a step in the right direction.

# Conclusion

This is an ok paper, but needs many improvements:

- Error bars for experiments
- Comparisons to previously published results
- It would greatly benefit from running this method with close to SOTA architecture, for example DenseNet for CIFAR
- Discussions on the properties of the proposed operation
- Derivations for batch-norm

## EDIT: Update

The paper was improved. In particular I like the added explanation of the batch-norm and some improved explanation and phrasing.

Nevertheless, the experimentation remains weak both compared to the state of the art and previously published work (e.g. CIFAR reported in Trabelsi et al., 2017 and Gaudet & Maida, 2017).

Therefore, I increase the score by one point, as this work is just very slightly above the acceptance threshold right now.

---

> ### Author Response · Authors · 2020-11-23
> **Response to Reviewer 1**
>
> - Regarding error bars, unfortunately we did not have time to run the experiments enough, but we saw from a few runs that the results stayed close. Our main goal was to show we captured the parts of the algebras that treated multi-dimensional data as a single entity, while at least achieving similar performance.
> - Regarding comparisons to previously published results, we tried to include all CNN experiments that were done in the Deep Complex and Deep Quaternion papers as well as the colorization experiment as providing good evidence that our work captured the forced local relations (each output dimension being a unique linear combination of all dimensions and a shared weight set) that give the networks their power.
> - Regarding SOTA architectures, we have the codebase ready to publish on release of this paper where doing this would be very fast, but we did not have time in the rebuttal period to perform the suggested additional experiments. Our goal was to do a comparison against the recent CNN work for Complex and Quaternion papers where only basic ResNets were used.
> - We revised the paper to emphasize that our main goal as to capture the aspects of the algebra that we proposed are important, the forced local relations. Our experiments reveal that other aspects of the complex/hyper-complex algebras are immaterial.
> - We have added the derivation for batch-norm in Section 3.3.
> - Regarding non SOTA for DSTL, the SOTA model was an ensemble model where different models were trained on different classes. We only needed to establish the effectiveness of vector map convolution in comparison to real-avlued convolutions to make a point about high dimensionality data.
> - Regarding the effcets of L, we have added a new experiment under the CAE Section 4 where we freeze L. We included the results in Table 2 where we see allowing L to be learned gives a slight performance increase.

---

### Decision · Program_Chairs · 2021-01-07
**Final Decision**

**Decision:**

Reject

**Comment:**

While the updated version of this manuscript did motivate one reviewer to give the paper a marginal accept rating, all other reviewers really felt that the paper could use more work along the lines of their suggestions. The aggregate view of the reviewers is just not positive enough at this time to warrant an accept recommendation by the AC at this time. The work does seem to have promise and the authors are encouraged to continue to improve the paper for another round of peer review elsewhere.